# Role of Residence Area on Diet Diversity and Micronutrient Intake Adequacy in Urban and Rural Costa Rican Adolescents

**DOI:** 10.3390/nu14235093

**Published:** 2022-12-01

**Authors:** Rafael Monge-Rojas, Rulamán Vargas-Quesada, Georgina Gómez

**Affiliations:** 1Nutrition and Health Unit, Costa Rican Institute for Research and Education on Nutrition and Health (INCIENSA), Ministry of Health, Tres Ríos 4-2250, Costa Rica; 2Biochemistry Department, School of Medicine, University of Costa Rica, San José 11501-2060, Costa Rica

**Keywords:** dietary diversity, nutritional adequacy, adolescents, Costa Rica

## Abstract

Dietary diversity might be essential to meet nutritional demands during adolescence. Diet diversity among 818 urban and rural Costa Rican adolescents aged 13–18 years was studied using the Minimum Dietary Diversity Score for Women. The Nutrient Adequacy Ratio (NAR) was calculated for 11 nutrients to estimate the nutrient adequacy of the diet. A NAR < 0.7 was considered inadequate for micronutrient intake. The optimal Diet Diversity Score (DDS) cut-off point for this study was 4, established using receiver-operating characteristic curves. The mean DDS for the overall sample was 4.17 ± 1.43, although DDS was significantly higher in adolescents from rural vs. urban areas (4.33 ± 1.43 vs. 4.00 ± 1.42, *p*-value = 0.001). The odds of having a diverse diet were 62% higher in rural vs. urban adolescents. Overall, 80–95% of adolescents reached a NAR ≥ 0.70 for 8 nutrients except for calcium, zinc, and vitamin A. The residence area plays a key role in adolescent dietary diversity. Although overall DDS was low, foods that make up the rural adolescent diet were nutritionally dense enough to satisfy the EAR for most micronutrients. A high DDS is not necessarily required for the diet to meet most micronutrient demands in adolescence. Improved dietary adequacy of vitamin A, zinc, and calcium is required due to the importance of these micronutrients in maintaining optimal health.

## 1. Introduction

The need for energy and nutrients increases significantly in adolescence to support the rapid rate of growth and development [1]. During this period of life, achieving an adequate nutrient intake is crucial; however, there is evidence that dietary quality declines from childhood to adolescence [2,3,4], and that adolescents, as a group, have poor eating habits that do not meet dietary recommendations for micronutrient intake [5]. Evidence suggests that healthy dietary habits established during adolescence continue into adulthood [6,7,8,9]; hence, adolescence has been identified as the best time to achieve dietary modifications aimed at enhancing health-conscious dietary habits [6,9,10]. Assessing the quality of the adolescent diet is necessary to prevent deficiencies that may limit optimal growth and development and increase the risk of developing cardiometabolic diseases in adulthood.

Several food-based and nutrient-based dietary quality indices have been developed in Europe and North America to assess the overall healthfulness of dietary intake in adolescents; however, these indices require intensive dietary evaluation, complex analysis, and a significant burden of resources [11]. Nevertheless, among dietary indices, the Dietary Diversity Score (DDS) has been identified at the individual level as a promising measurement tool, particularly for use in low-income countries (LICs) and middle-income countries (MICs) [12], because of its simplicity of implementation and low financial support required.

Food diversity has long been recognized as a key element of high-quality diets, based on the premise that no single food can provide the adequate amount of nutrients needed to maintain optimal health [12]; therefore, eating a variety of foods has been a longstanding recommendation in the dietary guidelines of various countries [13]. Lack of dietary diversity is usual among underserved adolescents from LICs and MICs, mainly because their diets are largely based on starchy staples, with little or no animal products, and few fresh fruits and vegetables [14,15,16].

Dietary diversity may be linked to four dietary archetypes arising from socio-ecological traps in the food systems prevalent in various countries, even in rural and urban areas of the same country [17,18]. The Traditional diet archetype is characterized by providing a healthy intake of various nutrients, whereas the Mixed diet archetype represents ample access to a large number of traditional and Western foods. The Undernourishing diet archetype is characterized by inadequate quality and potentially inadequate quantity of food, while the Overnourishing diet archetype tends to replace the traditional diet with a Western diet, which has an insufficient quantity of micronutrients rich and nutritious foods [17].

Due to the lack of a consensual definition of dietary diversity, several DDSs have been proposed. A recent systematic scoping review showed that many DDSs vary in terms of format (e.g., food item-based, food group-based, dietary guidelines-based, functional diversity-based, and others) and the dietary assessment of food intake (on a single 24-h recall or Food Frequency Questionnaire). However, most are consistent on the concept of counting foods or food groups consumed over a period [12].

Studies conducted on adolescents of LICs and MICs show that the DDS can range from 3.35 points [14] to 6.25 points [19]; nevertheless, these values are not comparable since the methodology to determine DDSs varies considerably from one study to another. What is consistent is that the higher the DDS, the more sufficient the diet is to meet micronutrient requirements, and that rural adolescents have a lower dietary diversity than urban ones [20,21], except in rural adolescents from India, where the DDS is equal to 8 points [22]. Nevertheless, this study has some relevant methodological aspects that make it notably different from other studies carried out in rural adolescents. First, foods were categorized into thirteen groups, and second, a simple count of food groups was made to calculate individual DDS, which ranged from 1 to 13.

To avoid so much variability in studying dietary diversity, it is necessary to establish a simple methodology that suits the context and food culture of a country or large geographic area. The Minimum Dietary Diversity for Women (MDD-W) indicator was developed by the Food and Agriculture Organization (FAO) of the United Nations and partners [23] to fill the need for a simple food-based indicator for measuring dietary diversity and micronutrient adequacy as key dimensions of diet quality in women of reproductive age. However, this indicator has also been used to assess the adequacy of micronutrient intake in children [24], adult men [25,26], and adolescent boys [25], evidencing its applicability in other population groups.

According to the MDD-W, women who consume 15 g or more from at least five out of ten defined food groups over a 24-h recall period are classified as having a minimally adequate DDS to meet the recommended intake of 11 important micronutrients (vitamin A, thiamine, riboflavin, niacin, vitamin B6, folate, vitamin B12, vitamin C, calcium, iron, and zinc). However, several studies [25,27,28] have pointed out the need to analyze the sensitivity and specificity of different DDS cut-off points to define which is most appropriate for reducing the risk of over-identifying population groups with nutritionally inadequate diets. Under this premise, other studies have established different DDS cut-off points; for example, ≥4 for Ugandan adolescents [15] and ≥5 for Mexican adolescents [25].

The aims of the current study were: (1) to determine the DDS cut-off point for the Costa Rican adolescent diet, (2) to compare dietary diversity between urban and rural adolescents, and (3) to estimate the proportion of adolescents that do not meet micronutrient dietary recommendations according to area of residence.

## 2. Materials and Methods

### 2.1. Study Population and Setting

Data came from a cross-sectional sample of adolescents (13–18 years old; 7–11th graders) enrolled in 18 schools (10 from urban areas, and 8 from rural areas) in the province of San José, Costa Rica, in 2017. Most Costa Rican adolescents (80%) are enrolled in school [29] and San José has the highest adolescent concentration (30%) in the country [30].

School selection and sample size were determined assuming a sampling error for a population proportion with correction for a finite population [31] following three phases: (1) Schools were selected using a proportional-size probability method to represent urban and rural municipalities within the province [32]. (2) Ten classrooms (two from each grade) were selected in each school using simple random sampling. In each classroom, all students were invited to participate in the study, provided an informed assent, and asked to obtain a signed informed consent from their parents if they were interested in participating in the study. (3) Among those who returned signed informed assent and informed consent forms, students were randomly selected to participate in the study. Close to 5% of the initial sample chose not to participate in the study before it started. The final study sample was 818 adolescents aged 13 to 18 years.

Data were collected during school hours in a private classroom. The Bioethics Committee of the Costa Rican Institute for Research and Education on Nutrition and Health (INCIENSA) approved all study protocols, and all guidelines for human subject research were strictly followed.

### 2.2. Sociodemographic Variables

A paper-based questionnaire was used to collect data on sex, age, residence area, socioeconomic status (SES), and nutritional status.

### 2.3. Anthropometric Assessment

Participants’ height and weight were determined by trained nutritionists following the methodology described by Preedy [33]. Body mass index (BMI) values were calculated, and adolescent nutritional status was determined using the BMI-for-age Z score recommended by WHO [34]: <−2: underweight; ≥−2 and <+1: healthy weight (eutrophy); ≥+1 and <+2: overweight, and ≥+2: obese. For data analysis, nutritional status was dichotomized as non-overweight (underweight and eutrophy) and overweight/obesity (overweight and obesity).

### 2.4. Dietary Intake Assessment

Dietary intake data were collected via 3-day food records completed by the participants in real time and reviewed by nutritionists. Participants were asked to complete the 3-day food records on two weekdays (Monday, Tuesday, Thursday, or Friday) and one weekend day (Saturday or Sunday). Half of the participants were randomly selected to record the foods and drinks that they consumed on Thursday, Friday, and Saturday, and the rest were asked to record their intake on Sunday, Monday, and Tuesday. Data were collected during nine months of the school year (February-November), reflecting seasonal variations for Costa Rica: rainy season (May-November) and dry season (December–April). The goal was to ensure that the data captured daily and seasonal variability in food consumption.

At each school, six trained nutritionists provided intake forms to the participants and taught them how to complete accurate food records by writing down detailed descriptions of what they ate and drank from the time they woke up in the morning to the time they went to bed at night, for three consecutive days. Adolescents were asked to write down the brand names of foods when appropriate, methods of preparation, and recipes for all the dishes and drinks whenever possible. The nutritionists taught the participants how to estimate the portion sizes of the foods and drinks they consumed using an established portion-size manual developed for Costa Rica [35]. The manual includes photographs and diagrams of typical local foods and their preparations, including 3–6 different portion sizes. Adolescents were also instructed to report portion size using measurements based on household utensils or volume and mass units.

Given the challenges with incomplete and inaccurate data when recording self-reported dietary data in young populations and specific demographic groups [36], the completed 3-day food records were thoroughly reviewed by the nutritionists by conducting one-on-one reviews with each participant during school hours. At this interview, the nutritionists inquired about commonly missed items or ingredients (i.e., added sweeteners, added fats, candies, beverages), added details about the kind of food or drinks that were consumed, verified or added any omitted portion sizes, and clarified any illegible items. The nutritionists used food models, fresh foods, and different utensils to verify serving and portion sizes. Only the food intake recorded by the adolescents on the second day (Monday or Friday) was used to determine the dietary diversity score. The second day of the 3-day food record was chosen because it was a weekday for the whole sample, making sure to select a day of habitual adolescent dietary intake. Meanwhile, the first day (Thursday or Sunday) or the third one (Saturday or Tuesday) would have been a weekend day at least for half the sample. This might induce bias in the dietary intake analysis since food consumption and diet quality differ between weekdays and weekends [37,38]. To corroborate this premise, a post-hoc sensitivity analysis was performed using the dietary data from the first day of the 3-day food records for all the analyses performed with data from day 2. This showed important differences in the results obtained when comparing the first and second days of the food record, which confirmed that including a weekend day modifies the results negatively (Appendix A, Appendix A).

### 2.5. Diet Diversity Score 

The DDS was assessed using the MDD-W of reproductive age developed by FAO in 2016 as a valid indicator of micronutrient adequacy in women of childbearing age [23]. This is a proposal of a single indicator to assess dietary quality that has been used in populations different from the one for which it was originally designed, such as adult men [25,26] and adolescent boys [15,25].

The DDS was calculated at the individual level based on the Women’s Dietary Diversity Score Project food group classification [23]; the first step involved adding the total of food groups reported as consumed on the second day of the 3-day food records (Friday or Monday). This was done because the established methodology by FAO to analyze the MDD-W should be done based only on a single 24-h period over one day and night [23].

To determine dietary diversity, a score was created based on the consumption of at least 15 g/day of the following 10 food groups: (1) starchy staples (grains, roots and tubers, and plantains), (2) flesh foods (meat, poultry, and fish), (3) dark green leafy vegetables, (4) other vitamin A-rich fruits and vegetables (other vitamin A-rich F&V), (5) other vegetables, (6) other fruits, (7) pulses (beans, peas, and lentils), (8) milk and milk products, (9) eggs, and (10) nuts and seeds. One point was assigned if dietary intake of a food group was ≥15 g/day and 0 points if intake was <15 g/day; thus, the score ranged from 0 to 10 points. Higher scores would indicate a diet with greater diversity, as more food groups were eaten in amounts greater than or equal to 15 g/d. DDS analyses were performed by sex, residence area, socioeconomic status (SES), and nutritional status.

### 2.6. Nutrient Adequacy

To estimate the nutrient adequacy of the diet, a Nutrient Adequacy Ratio (NAR) was calculated for the intake of 11 nutrients. The NAR for a given nutrient is the ratio of a subject’s intake to the Estimated Average Requirement (EAR) for the adolescent’s sex and age category, according to the guidelines of the National Academy of Medicine of the United States [39].

The EAR is the appropriate dietary reference intake to use when assessing the adequacy of group intakes [40]. A NAR value less than 70% (NAR < 0.7) was considered indicative of inadequate nutrient intake, as suggested by Oldewage-Theron [41]. As an overall measure of nutrient adequacy, the Mean Adequacy Ratio (MAR) was calculated as has been previously described [42].
$$Mean\;Adequacy\;Ratio\;\left(MAR\right)=\frac{\sum NAR\;\left(each\;truncated\;at\;1\right)}{Number\;of\;nutrients}$$

*NAR* was truncated at 1, so a nutrient with a high NAR could not compensate for a nutrient with a low NAR. MAR > 0.7 has been considered as the “gold standard” of nutritionally adequate intake [28].

### 2.7. Diet Diversity Score Cut-Off

According to the FAO methodology, the diet can be considered diverse if 5 or more food groups are consumed, and non-diverse if the consumption is less than 5 food groups [23]. However, the 5-food-group cut-off has been questioned and several studies have recommended determining the cut-off point according to the study population [25,27,28,41] to optimize balance among sensitivity, specificity, and positive predictive values for MAR [27]. In consequence, receiver-operating characteristic (ROC) curves were generated to obtain the adequate DDS cut-off for this study. As shown in Figure 1, different DDS cut-off points were tested for sensitivity and specificity versus different definitions of a nutritionally adequate diet, with MAR ranging from 0.65 to 0.75.

An optimal DDS cut-off point can identify as many nutritionally inadequate diets as possible (high sensitivity) without losing too much ability to identify nutritionally adequate diets (specificity). Therefore, nutritionally inadequate diets, defined here as MAR and DDS below a cut-off point, were defined as true positives. Nutritionally adequate diets, with MAR and DDS above a cut-off point, were defined as true negatives [28]. Figure 1 shows that the sensitivity and specificity were slightly influenced by changes in the MAR cut-off points, so we used a MAR of 0.70 as the cut-off point for a nutritionally adequate diet.

A MAR cut-off point equal to 0.70 and a DDS cut-off point equal to 4 resulted in 71.5% specificity and 78.0% sensitivity. If the DDS was increased to 5, specificity decreased to 43.5%, while sensitivity increased to 91.5%. Thus, a DDS cut-off point of 4 is, in this study, the one that resulted in higher than 70% values for both specificity and sensitivity. This cut-off point is considered robust, since the Youden J-index value (0.495) was the closest to 1, compared to the values observed while analyzing the other DDS cut-off points. Also, this DDS cut-off point offers a balance between a high negative predictive value and a modest positive predictive value for MAR (0.97 and 0.23, respectively), reducing the risk of over-identifying adolescents with nutritionally inadequate diets [43].

### 2.8. Statistical Analyses

Data were reported as means ± standard deviations (SD) for continuous variables, and frequencies (%) for categorical variables. The between-groups comparisons (age, sex, SES, and nutritional status) in the proportions of adolescents according to residence area were tested using the Wilcoxon test, or the chi-square test with Bonferroni post hoc test when required. The DDS was stratified by sex, SES, and nutritional status, and the difference between groups was tested with Student’s t or ANOVA tests as required. The odds of having a diverse diet according to sociodemographic and nutritional status variables were determined by a bivariate and multivariate logistic regression analysis. All tests were two-tailed, and *p*-values < 0.05 were considered statistically significant.

Consumption of food groups and nutrient intake were stratified by DDS (<4 or ≥4), and differences between groups were tested using the Wilcoxon test. Intakes were adjusted for 1000 kcal to assess the dietary diversity effect (DDS < 4 vs. DDS ≥ 4) on macronutrient and micronutrient intake, independently of diet quantity. NAR and MAR were stratified by diet diversity and comparisons were made using the Wilcoxon test. Spearman’s rank correlation coefficients were used to determine the association between the DDS and NAR/MAR scores. Finally, a chi-square test was used to estimate significant differences in the distribution of participants with NAR/MAR ≥ 0.70 according to DDS and residence area.

Data were analyzed using STATA software (version 14.1, 2015, College Station, TX, USA). Epi Info™ software (version 3.5.4, 2008) was used to process data from the 3-day food records, with information from the database of the School of Human Nutrition of the University of Costa Rica [44]. This database contains 1655 items from the following sources: 1307 foods from USDA, 80 foods from the food composition tables of the Institute of Nutrition for Central America and Panama (INCAP), 254 foods from recipes commonly consumed in Costa Rica (e.g., frescos, pastries, bread, cookies, and desserts), 13 foods from the Costa Rican mandatory fortification food group and updated information until 2017 (year in which the study data were collected).

## 3. Results

### 3.1. General Characteristics According to Residence Area

The mean age of the study sample was 15.0 (±1.7) years, with 63.6% girls, 50.2% urban, and 32.6% overweight/obese adolescents (Table 1). There was no significant difference between adolescents from urban and rural areas regarding age, sex, and overweight/obesity status (*p* > 0.05). Compared to urban adolescents, a higher proportion (66.9%) of rural adolescents was classified as low SES (*p* < 0.0001). Likewise, a higher proportion of urban adolescents was enrolled in private schools compared to rural adolescents.

### 3.2. Diet Diversity Score According to General Characteristics

The mean DDS for the overall sample was 4.17 ± 1.43 points out of 10.0 possible maximum score, with 66.5% of participants reaching 4 or more points, which is the criterion for a diverse diet (Table 2). DDS was higher in participants from rural vs. urban areas (4.33 ± 1.43 vs. 4.00 ± 1.42) (*p* = 0.001), and 72.0% of rural adolescents reached the criterion for a diverse diet, compared with 61.1% for urban adolescents (*p* = 0.001). No significant differences were found when analyzing the DDS by sex, SES, or nutritional status.

### 3.3. Consumption of Food Groups

As expected, adolescents with a diverse diet (DDS ≥ 4) showed a significantly higher consumption of all food groups than participants with a non-diverse diet (DDS < 4) (*p* < 0.05) (Table 3). Adolescents from rural areas had higher consumptions of starchy staples (Δ = 49.0 g/d; *p* < 0.0001); pulses (Δ = 31.0 g/d; *p* < 0.0001); other vegetables (Δ = 19.3 g/d; *p* <0.0001); eggs (Δ = 4.2 g/d; *p* = 0.029); and dark green leafy vegetables (Δ = 3.7 g/d; *p* = 0.016). In contrast, urban adolescents had a higher dietary intake of milk and milk products (Δ = 27.7 g/d; *p* = 0.010) (Table 3). There were no differences in the consumption of flesh foods, other fruits, other vitamin A-rich F&V, and nuts and seeds, between urban and rural adolescents (*p* > 0.05).

Figure 2 shows the proportion of adolescents who consumed the food groups that presented significant differences between urban and rural areas. The proportion of urban and rural adolescents that consumed the starchy food group was similar, possibly because the largest proportion (35%) of this food group corresponds to rice (data not shown), which is one of the main staple foods of the Costa Rican diet.

A higher proportion of rural vs. urban adolescents consumed at least 15 g of the following food groups: pulses (73.5% vs. 50.6%, *p* < 0.0001); other vegetables (54.1% vs. 39.2%, *p* < 0.0001); eggs (32.2% vs. 24.6%; *p* = 0.016); and dark green leafy vegetables (6.1% vs. 2.7%; *p* = 0.016). On the contrary, a higher proportion of urban vs. rural adolescents consumed 15 g or more of milk and milk products (57.4% vs. 47.7%; *p* = 0.005).

Table 4 shows a bivariate and multivariate logistic regression analysis with the sociodemographic variables related to having a diverse diet. According to the multivariate model, the odds of having a diverse diet are 62% higher in rural adolescents, after adjusting for sex, age, SES, and nutritional status (OR 1.62, 95% CI: 1.19–2.20, *p*-value = 0.002).

### 3.4. Nutrient Adequacy Ratio and Correlation with Diet Diversity Score

Table 5 shows NAR values for each nutrient. Mean NARs for calcium and vitamin A are below 70% of the EAR (NAR < 0.70), independently of diet diversity. Adolescents with a diverse diet (DDS ≥ 4) reported significantly higher NAR values compared to participants with a non-diverse diet (DDS < 4) for the 11 micronutrients assessed. MAR values were, as expected, significantly higher in the diverse diet group. For the whole sample, MAR and all NARs correlated positively and significantly with the DDS, showing an increase in the micronutrient adequacy ratio for all these vitamins and minerals as DDS increased. Higher correlations were obtained for zinc (0.354), calcium (0.383), and MAR (0.433).

Table 6 shows the proportion of participants reaching a NAR ≥ 0.70, according to diet diversity and residence area. Overall, less than 70% of the adolescents reached a NAR ≥ 0.70 for calcium, zinc, and vitamin A. As expected, for all nutrients the proportion of adolescents reaching a NAR ≥ 0.70 with a diverse diet (DDS ≥ 4) was significantly higher than for adolescents with a non-diverse diet (DDS < 4). Regarding residence area, a significantly higher proportion of rural vs. urban adolescents reached a NAR ≥ 0.70 for zinc (69.0% vs. 61.6%; *p* = 0.025), vitamin C (87.0% vs. 81.8%; *p* = 0.040), and vitamin B6 (87.0% vs. 79.3%; *p* = 0.003). In contrast, this proportion was higher in urban adolescents only for calcium (29.9% vs. 22.4%; *p* = 0.014). No significant differences were observed between rural and urban adolescents for other nutrients/MAR.

## 4. Discussion

This study shows that the DDS cut-off point for urban and rural Costa Rican adolescents was similar to the one for Ugandan adolescents (≥4) [15], and lower than the one for Mexican adolescents (≥5) [25]. The cut-off point of 4 was determined through sensitivity and specificity analyses, which are inversely related. For nutrition promotion and prevention of nutrient inadequacy, high sensitivity is desirable to accurately identify adolescents with inadequate micronutrient intake, as false positives pose no significant risk in this case [28]. However, the determination of a DDS cut-off point with a balance between sensitivity and specificity is essential to strengthening the analysis of dietary diversity, since failure to do so may increase the risk of overestimating the proportion of adolescents with inadequate micronutrient intakes. For example, in this study, a DDS cut-off point of 5 would have given a specificity of 43.5%, with a false positive rate of 56.5% (28.0% higher than with a DDS cut-off point of 4), which would predict a higher proportion of adolescents with inadequate micronutrient intakes.

In contrast to the evidence from LICs, where diet diversity is lower in rural areas, in Costa Rica (a MIC) diet diversity is higher in rural areas. Among the main causes of low diet diversity in LICs are dependence on home-grown staples and legumes, and limited access (economic and physical) to other key healthy foods that are available at the local markets—typical of the Undernourishing diet archetype [17].

In Costa Rica, food availability and access in rural areas (particularly in San José, where this study was conducted) are not very different from the urban areas of the province [45]. The conceptualization of ‘rural’ in Costa Rica may differ from what is established in LICs. Currently, the diversification of the rural productive structure no longer depends exclusively on agricultural activities. Instead, there has been a significant growth of service activities, which are increasingly not associated with agriculture. In addition, there is a more functional integration of rural and urban settings, which can be characterized as a process of counter-urbanization. This process has been facilitated by the development of new communication technologies and an increased income in rural populations, which help bridge the gap between the ‘modern’ and rural worlds [46].

However, rural areas still retain certain unique qualities such as family food patterns. It has been reported previously that the rural adolescent diet tends to be more traditional than the urban adolescent diet. The traditional diet in rural adolescents is characterized by adequate quantities, diversity, and quality of food to sustain health and nutrition [45]. This can help explain why rural adolescents have 62% odds of having a more diverse diet than their urban counterparts. The diet diversity observed in rural areas suggests that the Traditional diet archetype prevails in those areas.

Although the overall DDS was low (4.17 ± 1.43 out of 10 points) and similar to Uganda’s, there was a low proportion of rural and urban adolescents with inadequate intakes. Hence, even if a low DDS is an indicator of a monotonous diet [12], it does not necessarily lead to an inadequate micronutrient intake, as adequacy will depend on the foods that fall within that monotony. Beans and rice, which are the staple foods of the Costa Rican diet, provide a great variety of nutrients. Pulses are natural sources of thiamine, riboflavin, niacin, vitamin B6, folate, iron, and zinc [47], while fortified white rice provides thiamine, niacin, folic acid, and zinc [48].

In contrast to urban adolescents, a large proportion of rural adolescents consumes at least 15 g/day of pulses, dark green leafy vegetables, other vegetables, and affordable protein sources such as eggs, contributing to an improved DDS in rural areas —even if slightly above the minimum diet diversity set for this study. This finding suggests that neither a high DDS nor a high family income is necessarily required for the diet to meet the majority of micronutrient demands in adolescence.

The lower DDS found in urban areas may be reflecting a more established nutritional transition. It has been thoroughly demonstrated that rapid urbanization and high family income are critical drivers of this transition [49,50,51,52]. Nutritional transition is characterized by the progressive shift from a traditional diet to one rich in animal products and low in pulses [53], corresponding to the Overnourishing diet archetype [17], and evidenced in the urban adolescent diet. Therefore, increased consumption of costly foods (e.g., milk and milk products) at the expense of more affordable choices (pulses and eggs) does not guarantee a higher dietary diversity.

Although the prevalence of inadequate micronutrient intakes was low for most micronutrients, interventions need to be designed to increase the consumption of vitamin A-rich fruits and vegetables, eggs, and flesh foods—particularly in urban adolescents—to improve the dietary adequacy of vitamin A and zinc. Likewise, promoting the consumption of calcium-rich dairy products is urgently required, since the intake of this mineral is critical for both urban and rural adolescents.

We acknowledge the limitations of our study. Although the MDD-W was not designed for men, other studies conducted in adult and adolescent males [15,25,26] have demonstrated its usefulness in determining the adequacy of micronutrient intake; hence, its use in males should be considered and analyzed with greater deference. The MDD-W is a simple and affordable tool requiring very straightforward analysis, and the determination of food consumption does not involve complex dietary assessment methods.

Our study also has strengths that must be emphasized. First, we conducted the DDS analysis using a specific cutoff point for the population group, providing a balance between a high negative predictive value and a modest positive predictive value for MAR, and reducing the risk of over-identifying adolescents with nutritionally inadequate diets. Second, diet evaluation through intake records is considered the gold standard among other dietary assessment instruments [54]. Finally, while participants may not be representative of Costa Rica as a whole, the province of San Jose is where most of the national adolescent population (30%) is concentrated [30].

## 5. Conclusions

Our study shows that the area of residence plays a key role in adolescent dietary diversity. Although overall DDS was low, the foods that make up the rural adolescent diet were nutritionally dense enough to satisfy the EAR for most micronutrients. Therefore, our study suggests that a high DDS is not required to achieve ≥ 0.70 EAR intake for most micronutrients as long as the foods included in the diet have a high micronutrient density either naturally or through fortification. However, greater consumption of vitamin A-rich fruits and vegetables, eggs, and meats is required—particularly in urban adolescents—to improve the dietary adequacy of vitamin A and zinc. In addition, increased consumption of calcium-rich food sources is urgently required for all adolescents because inadequate intake of this mineral is notoriously high.

## Figures and Tables

**Figure 1 nutrients-14-05093-f001:**
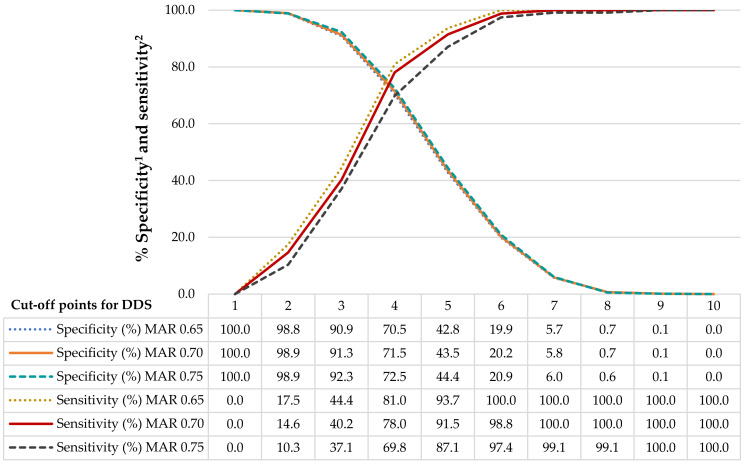
Sensitivity and specificity of different DDS cut-off points with MAR cut-off points ranging from 0.65 to 0.75. ^1^ Specificity: identifies nutritionally appropriate diets as adequate. ^2^ Sensitivity: identifies nutritionally inappropriate diets as inadequate. DDS: Diet Diversity Score; MAR: Mean Adequacy Ratio.

**Figure 2 nutrients-14-05093-f002:**
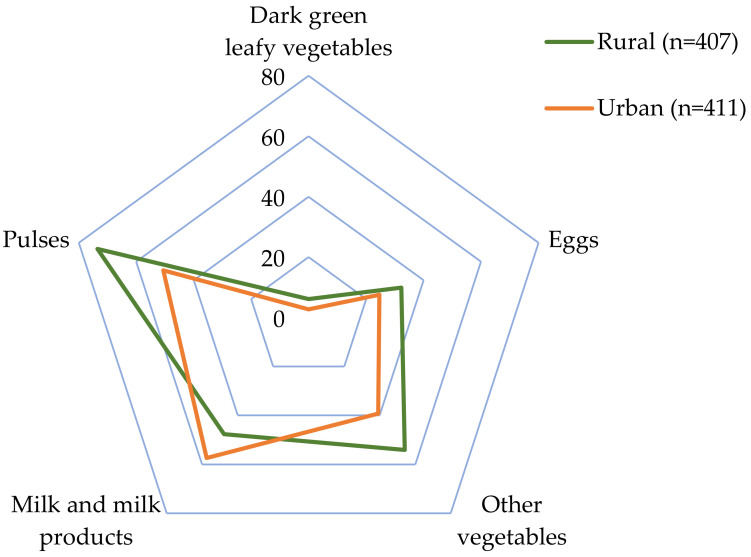
Proportion of participants consuming at least 15 g of each food group over one-day food record, according to residence area.

**Table 1 nutrients-14-05093-t001:** General characteristics of Costa Rican adolescents, according to residence area.

Characteristics	Overall ^1^ (*n* = 818)	Residence Area		
		Urban (*n* = 411)	Rural (*n* = 407)	*p*-Value ^2^
Age (y)	15.0 ± 1.7	14.9 ± 1.7	15.1 ± 1.7	0.208
Sex				
Female	520 (63.6)	259 (49.8) ^a^	261 (50.2) ^a^	0.741
Male	298 (36.4)	152 (51.0) ^a^	146 (49.0) ^a^	
Socioeconomic status				
Low	263 (32.2)	87 (33.1) ^a^	179 (66.9) ^b^	<0.0001
Middle	325 (39.7)	158 (48.6) ^a^	167 (51.4) ^a^	
High	230 (28.1)	166 (72.2) ^a^	64 (27.8) ^b^	
Nutritional status				
Non-overweight	551 (67.4)	279 (50.6) ^a^	272 (49.4) ^a^	0.748
Overweight/obesity	267 (32.6)	132 (49.4) ^a^	135 (50.6) ^a^	

^1^ Values are means ± SD or frequencies (%) unless otherwise indicated. ^a,b^ Labeled frequencies in a row without a common letter differ (*p* < 0.05). ^2^
*p*-values < 0.05 are statistically significant and were determined using the Wilcoxon test, or the chi-square test with Bonferroni post hoc test when required. *p*-values compare either means of age or proportions in categories of sex, socioeconomic status, and nutritional status between categories of residence area (urban/rural).

**Table 2 nutrients-14-05093-t002:** Diet Diversity Score (DDS) and proportion of participants reaching the minimum DDS ^1^ (*n* = 818).

Characteristic	Diet Diversity Score (DDS)			Participants Reaching the Minimum DDS	
	Mean	SD	*p*-Value ^2^	*n* (%)	*p*-Value ^2^
Overall	4.17	1.43		544 (66.5)	
Sex					
Female	4.18	1.43	0.679	349 (67.1)	0.624
Male	4.14	1.44		195 (65.4)	
Residence area					
Urban	4.00	1.42	0.001	251 (61.1)	0.001
Rural	4.33	1.43		293 (72.0)	
Socioeconomic status					
Low	4.28	1.44	0.303	183 (69.6)	0.420
Middle	4.11	1.39		213 (65.5)	
High	4.11	1.48		148 (64.4)	
Nutritional status					
Non overweight	4.15	1.43	0.680	367 (66.6)	0.929
Overweight/obesity	4.19	1.43		177 (66.3)	

^1^ Consumption of at least 4 out of 10 food groups. ^2^
*p*-values < 0.05 are statistically significant and were determined using the chi-square, Student’s t, or ANOVA tests.

**Table 3 nutrients-14-05093-t003:** Food group consumption of Costa Rican adolescents, according to Diet Diversity Score (DDS) and residence area.

Food Group ^1^ (g/d)	Diet Diversity			Residence Area		
	DDS < 4 (*n* = 274)	DDS ≥ 4 (*n* = 544)	*p*-Value ^2^	Urban (*n* = 411)	Rural (*n* = 407)	*p*-Value ^2^
Starchy staples	357.1 ± 184.3	387.7 ± 197.9	0.031	353.1 ± 190.0	402.1 ± 196.8	<0.0001
Milk and milk products	91.1 ± 204.0	166.5 ± 207.1	<0.0001	155.1 ± 220.7	127.4 ± 195.8	0.010
Pulses	55.1 ± 108.2	106.9 ± 112.2	<0.0001	74.1 ± 105.9	105.1 ± 118.8	<0.0001
Flesh foods	56.5 ± 84.7	102.3 ± 94.5	<0.0001	91.3 ± 98.6	82.6 ± 88.6	0.245
Other fruits	25.4 ± 78.1	91.8 ± 145.0	<0.0001	73.4 ± 145.5	65.7 ± 109.2	0.816
Other vegetables	11.8 ± 36.3	62.3 ± 86.8	<0.0001	35.8 ± 69.4	55.1 ± 84.0	<0.0001
Eggs	7.0 ± 21.9	23.4 ± 37.9	<0.0001	15.9 ± 32.7	20.1 ± 35.7	0.029
Other vitamin A-rich F&V	5.9 ± 45.6	15.0 ± 57.3	<0.0001	14.3 ± 64.1	9.7 ± 40.9	0.405
Dark green leafy vegetables	0.9 ± 14.5	6.7 ± 31.4	<0.0001	2.9 ± 21.3	6.6 ± 31.8	0.016
Nuts and seeds	0.3 ± 3.4	2.8 ± 20.0	0.003	1.8 ± 14.2	2.1 ± 18.5	0.467

^1^ Values are means ± SD. ^2^
*p*-values < 0.05 are statistically significant and were determined using the Wilcoxon test. DDS: Diet Diversity Score; F&V: Fruits and vegetables.

**Table 4 nutrients-14-05093-t004:** Bivariate and multivariate logistic regression analysis: Sociodemographic variables related to having a diverse diet, according to Diet Diversity Score (DDS) (*n* = 818).

Variable	Bivariate Analysis			Multivariate Analysis ^1^		
	OR	95% CI	*p*-Value	Adj OR	95% CI	*p*-Value
Sex						
Women	1.08	0.80–1.46	0.624	1.08	0.80–1.46	0.622
Age	1.09	1.00–1.19	0.047	1.08	0.99–1.18	0.067
Residence area						
Rural	1.64	1.22–2.20	0.001	1.62	1.19–2.20	0.002
Socioeconomic status						
Middle	0.83	0.59–1.18	0.299	0.91	0.64–1.30	0.617
High	0.79	0.54–1.15	0.217	0.98	0.66–1.46	0.922
Nutritional status						
Overweight/obesity	0.99	0.72–1.34	0.929	0.99	0.73–1.36	0.989
Constant	-	-	-	0.46	0.12–1.84	0.275

^1^ Overall model test *p*-value = 0.020. Goodness-of-fit test *p*-value = 0.470. Correct predictions: 66.5%.

**Table 5 nutrients-14-05093-t005:** Mean Nutrient Adequacy Ratio (NAR), according to Diet Diversity Score (DDS).

Nutrients	Overall (*n* = 818)		DDS < 4 (*n* = 274)		DDS ≥ 4 (*n* = 544)			*r* ^1^	*p*-Value
	Mean	SD	Mean	SD	Mean	SD	*p*-Value ^2^		
Calcium	0.492	0.292	0.368	0.271	0.555	0.283	<0.0001	0.383	<0.0001
Iron	0.968	0.131	0.938	0.190	0.983	0.084	0.001	0.171	<0.0001
Zinc	0.788	0.249	0.666	0.284	0.849	0.204	<0.0001	0.354	<0.0001
Vitamin C	0.885	0.268	0.778	0.357	0.939	0.187	<0.0001	0.265	<0.0001
Thiamin	0.965	0.132	0.923	0.197	0.985	0.073	<0.0001	0.205	<0.0001
Riboflavin	0.940	0.173	0.869	0.248	0.976	0.102	<0.0001	0.276	<0.0001
Niacin	0.954	0.151	0.900	0.223	0.981	0.084	<0.0001	0.232	<0.0001
Vitamin B6	0.898	0.213	0.802	0.284	0.947	0.143	<0.0001	0.311	<0.0001
Folate equivalents	0.969	0.128	0.934	0.189	0.987	0.075	<0.0001	0.163	<0.0001
Cobalamin	0.957	0.159	0.908	0.234	0.982	0.093	<0.0001	0.215	<0.0001
Vitamin A	0.686	0.310	0.578	0.338	0.740	0.280	<0.0001	0.309	<0.0001
MAR	0.864	0.135	0.788	0.176	0.902	0.087	<0.0001	0.433	<0.0001

^1^ Spearman’s rank correlation coefficients (*r*) were calculated between each NAR/MAR value and the DDS for the whole sample. ^2^
*p*-values < 0.05 are statistically significant and were determined using the Wilcoxon test. NAR: Nutrient Adequacy Ratio; MAR: Mean Adequacy Ratio; DDS: Diet Diversity Score.

**Table 6 nutrients-14-05093-t006:** Proportion of participants reaching Nutrient Adequacy Ratio (NAR) ≥ 0.70, according to Diet Diversity Score (DDS) and residence area.

Nutrient	Overall ^1^ (*n* = 818)	DDS < 4 (*n* = 274)	DDS ≥ 4 (*n* = 544)		Urban (*n* = 411)	Rural (*n* = 407)	
	*n* (%)	*n* (%)	*n* (%)	*p*-Value ^2^	*n* (%)	*n* (%)	*p*-Value ^2^
Calcium	214 (26.2)	38 (13.9)	176 (32.4)	<0.0001	123 (29.9)	91 (22.4)	0.014
Iron	776 (94.9)	249 (90.9)	527 (96.9)	<0.0001	386 (93.9)	390 (95.8)	0.217
Zinc	534 (65.3)	127 (46.4)	407 (74.8)	<0.0001	253 (61.6)	281 (69.0)	0.025
Vitamin C	690 (84.4)	195 (71.2)	495 (91.0)	<0.0001	336 (81.8)	354 (87.0)	0.040
Thiamin	774 (94.6)	240 (87.6)	534 (98.2)	<0.0001	386 (93.9)	388 (95.3)	0.370
Riboflavin	743 (90.8)	218 (79.6)	525 (96.5)	<0.0001	368 (89.5)	375 (92.1)	0.198
Niacin	752 (91.9)	228 (83.2)	524 (96.3)	<0.0001	378 (92.0)	374 (92.0)	0.967
Vitamin B6	680 (83.1)	184 (67.2)	496 (91.2)	<0.0001	326 (79.3)	354 (87.0)	0.003
Folate equivalents	779 (95.2)	245 (89.4)	534 (98.2)	<0.0001	389 (94.7)	390 (95.8)	0.430
Cobalamin	762 (93.2)	236 (86.1)	526 (96.7)	<0.0001	386 (93.9)	376 (92.4)	0.385
Vitamin A	430 (52.6)	110 (40.2)	320 (58.8)	<0.0001	215 (52.1)	215 (52.8)	0.883
MAR	736 (90.0)	210 (76.6)	526 (96.7)	<0.0001	368 (89.5)	368 (90.4)	0.675

^1^ Values are frequencies (%) unless otherwise indicated. ^2^
*p*-values < 0.05 are statistically significant and were determined using the chi-square test. NAR: Nutrient Adequacy Ratio; MAR: Mean Adequacy Ratio; DDS: Diet Diversity Score.

## Data Availability

The data presented in this study are available on request to the corresponding author.

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
