# Peer review of "Role of Residence Area on Diet Diversity and Micronutrient Intake Adequacy in Urban and Rural Costa Rican Adolescents"

_nutrients, 2022, doi:10.3390/nu14235093_

Round 1

Reviewer 1 Report (Previous Reviewer 2)

I have read the authors' responses to my comments and suggestions and they have been very satisfactory. I have no further comments to make.   

Reviewer 2 Report (Previous Reviewer 1)

Thank you for addressing my earlier comments.

This manuscript is a resubmission of an earlier submission. The following is a list of the peer review reports and author responses from that submission.

Round 1

Reviewer 1 Report

The authors reported results on a study among adolescents that determined the cut-off point for their diet diversity score, compared dietary diversity between urban and rural dwellers, and proportions who did not meet dietary recommendations. Their results show that rural residents have better diet diversity that reflects consumption of food groups and nutrients which met recommendations compared to their urban counterparts.

The manuscript is well-written and the report is well-presented but I have one major concern:

Lines 167-168 and 177-178: Any reason for why you only chose one day as basis for analysis (DDS computation, etc)? One reason for collecting multiple food records is to determine habitual intake. Diversity in habitual intake (i.e., an average of the 3 days of diet records) would have been a more precise measure of diversity and intakes of food groups and nutrients. I'm wondering why you collected 3 consecutive days of food records but used only one of those 3 days when we know that there is daily variation in dietary intake. You already have the data that allows you to estimate habitual intake, which logically and practically, should be the basis for determining diversity in intake.

Minor issues:

1. Title of the article: The term "effect" is not appropriate for this article since this is a observational study/cross-sectional analysis, not an experimental study.

2. P-values of 0.000 are interpreted as highly significant but would be best presented as p < 0.0001. Please make this change in the text and tables.

3. Lines 180-181: Why 15 g (1/2 an ounce) across the board? Considering dietary guidelines, we will want certain food/food groups to be more in the diet compared to others (e.g., cereal grains are usually required in higher number of servings compared to vegetables, etc.). It will make more sense  so I was expecting that when scoring, the gram amount would vary from food group to food group.

4. Line 203: spelling of adequacy

Reviewer 2 Report

The authors of the study entitled “Effect of Residence Area on Diet Diversity and Micronutrient Intake Adequacy in Urban and Rural Costa Rican Adolescents” have carried out an exhaustive research work on the diet and nutrition of adolescents in Costa Rica, reporting on the areas to improve to achieve a healthier adolescent population. In my opinion, the possible conditioning variables have been well-chosen and the study was executed satisfactorily. However, I would like to share with the authors a number of questions and make some suggestions.

Major comments

1.    In relation to the Diet Diversity Score (DDS), I think it would have been interesting to divide the age of the adolescents into “pre-adolescents” and “older” adolescents (the terminology is not important here), since there could be significant differences between them. Thus, the age group among 13 and 15 years old is perhaps more dependent of the eating habits of their parents, while the age group among 16 and 18 begins to be more independent of the habits and norms established in their home, despite living in the same familiar environment in many cases.

2.    Tables

I have a series of doubts and suggestions about the statistics shown in the tables that I would like to share with the authors.

-       In tables 1, 2, 3, 5, and 6 the significance value was not given. That is, from what p-value are the data statistically significant? p < 0.05, p < 0.01, or p < 0.001?

-       Table 1 provides the p-value for each group belonging to the variables sex, socioeconomic status, and nutritional status when variables “urban” and “rural” are compared. However, only the value for the first group of each variable appears. For instance, in the case of sex, you only wrote the p-value in female, what about the male? Is it the same value for both sexes? I imagine so, but I think the authors should indicate it anyway.

-       In same table (table 1), the authors statistically compare the groups age, socioeconomic status, and nutritional status in relation to the residence area, giving the p-value. However, a statistical comparison between these groups and each residence area (urban and rural) is not provided. For example ¿is the amount of population with low, middle, and high incomes in the cities significantly different? And in the countryside? It is possible that this comparison is statistically of little value. Clarification on this would be nice. Otherwise, the authors should provide a post hoc analysis when there are more than two groups to compare, as in the case of socioeconomic status, as long as equal population variances are assumed.

Minor comments

1.  The title of figure 1 uses the expression “Specificity: identifies nutritionally adequate diets as adequate”. In mi opinion, the phrase does not sound very good, so the term “adequate” (the first one) should be replaced by another that makes the phrase sounds better like “appropriate”. Similarly, the term “inadequate” in the following tittle sentence “Sensitivity: identifies nutritionally inadequate diets as inadequate” should be replaced by another like “inappropriate”.

2.  The phrase “Mean NARs for calcium…independently of diet diversity” at the beginning of section 3.4 (lines 334-336) is not well-understood. It should be rewritten.

3.  I believe it is appropriate to provide, both in tables and figures, the corresponding meaning to all the abbreviations used, even if these are extensively used in the subject and/or they have already clarified through the text, since it is possible that the reader is interested in reading a table or figure before the text itself, so a clarification in this regard would speed up the reading and comprehension.

Round 2

Reviewer 1 Report

Concerns are satisfactorily answered.Thank you.

However, I suggest that the reason for why the second day of the 3-day food records was used (instead of the first or the 3rd day) be included in the manuscript. In addition, a brief statement about how FAO's DDS computations are based on just a 1-day food report/recall would improve readability among readers not familiar with FAO's DDS. These additional information would be most appropriate in section 2.5 of the manuscript.
